# Intersections of Children's Poetry, Popular Literature, and Mass Media: Fujimoto Giichi's Adaptation of *Holes in the Tin Roof like Stars* from Tomo Fusako's Poem to Radio Drama

Koji Toba

Faculty of Letters, Arts and Sciences, Waseda University, Tokyo 162-8644, Japan; toba@y.waseda.jp

**Abstract:** This paper investigates largely unexplored aspects of the postwar Japanese media industry by tracing the cross-media developments that bloomed from a single poem written by an elementary school girl. Tomo Fusako, a poor elementary school student, wrote the poem "Outage" in 1951 as part of her schoolwork. Tomo's teacher, Bessho Yasoji, selected Tomo's work to be published in an original poetry journal featuring children's writing. Her poems and essays were eventually reprinted in magazines, collected volumes, and even published in textbooks. In 1958, Fujimoto Giichi, an unknown university student at the time, adapted "Outage" into a radio drama and stage play. These works were then further adapted for TV dramas. Children's essays and poems made for attractive content for the publishing industry and the emerging fields of commercial radio and television media. Fujimoto himself became a famous television host, though it impeded his literary career. Examining Tomo and Fujimoto's relationship with literary production and media adaptation reveals a cultural world far removed from the literary establishment's (that is, the *bundan*'s) view of literature.

**Keywords:** children's poetry; children's essays; mass media; radio drama; TV drama; play; popular literature; Fujimoto Giichi

## 1. Introduction

Both in Japan and abroad, the study of modern Japanese literature began with the study of the so-called "Literary Masters" (*bungō*). The scope and methodology of modern literary studies, however, has gradually expanded beyond establishment writers, such as Natsume Sōseki and Mori Ōgai, who were previously the objects of academic research. Research in the English-speaking world also followed this trend in the decades following the end of World War II, progressing from studies on writers such as Kawabata Yasunari and Tanizaki Junichirō—who were perceived to embody some amorphous classical image of "Japan"—to studies of postwar writers such as Abe Kōbō and Ōe Kenzaburō. In even more recent years, literary research has swiftly and drastically expanded its critical reach. Critical interventions through emergent fields, such as minority literary studies, has led to a growing interest in Okinawan and Zainichi Korean writers. Film adaptations are now read in conjunction with printed texts, with a particular focus on so-called "literary films" (*bungei eiga*). Scholars have begun to look beyond national borders, with an example being Edward Mack's study of Japanese literature in Brazil, and others have begun to look beyond the border of "the text", such as Alexander Zahlten's critique of the "media mix" developed by Kadokawa Shoten in the business of literature and film (Zahlten 2017).

One often overlooked horizon of literary production is in the field of Japanese pedagogy. Modern Japanese school teachers developed a wide range of practices in their classrooms to encourage their pupils' compositions and poetry. Ōki Ken'ichirō taught a "genius" young student named Toyoda Masako. He published her essays and his own comments on them as *The Composition Class* (*Tsuzurikata kyōshitsu*) in 1937. Filmmaker Yamamoto Kajirō adapted the work—and, by extension, Toyoda's original literature—to a film the following year. The practice extended beyond Japan, as well. In Korea, directors

Choi In-gyu and Bang Hanjunmade the film *Tuition* (*Su-eop-ryo*) in 1940.[1] Works adapted from pedagogical approaches flourished in the postwar period beginning with *School of Echoes* (*Yamabiko gakkō*), released by the publisher Seidosha in 1951. The book collects works produced by the students of a young junior high school composition teacher from rural Yamagata Prefecture, Muchaku Seikyō. Muchaku's melding of pedagogical and literary practices inspired a variety of adaptations in postwar media (the magic lantern and film being two examples), igniting a nationwide movement to mimic his teaching and publication practices (Toba 2019). Many other films based on children's essays were produced throughout the 1950s, including Hisamatsu Seiji's *The Child Writers* (*Tsuzurikata kyōdai*) in 1958 and Imamura Shōhei's *The Diary of Sueko* (*Nianchan*) in 1959, which drew inspiration from the diary of a ten-year-old resident Korean girl.

In considering the relationship between modern Japanese literature and the media industry, this paper will focus on a radio drama that has not been previously examined, despite its proximity to these kinds of "composition films" (*tsuzurikata eiga*): Fujimoto Giichi's *Holes in the Tin Roof Like Stars* (*Totan no Ana wa Hoshi no yo*). This radio drama is based on a poem written by Tomo Fusako, an elementary school girl in Osaka. Fujimoto originally penned the script while he was a university student and he went on to become a Naoki-Prize-winning author. This paper traces the process through which a poem written by a poor but gifted elementary school girl gained wider attention. I intend to demonstrate how modern Japanese literature and Japanese language education have recognized literary talent at the margins. Exploring the reception of Fujimoto's scripts, moving from earlier radio dramas to his later theatrical works, reframes developments in postwar broadcasting and theater. In Fujimoto, we can see an example of a postwar writer whose identity was shaped through both literary and media experiences, particularly mass media, such as film and television.

**2. Making Masterpieces: How an Unknown Girl's Poetry Came to Appear in Textbooks**

In February 1951, Bessho Yasoji, a teacher at Minato Elementary School in Sakai, Osaka, observed a fourth-grade poetry class at an elementary school in K City (probably Kobe). Bessho received a printed collection of class poems and returned home determined to implement poetry sessions into his own class. Bessho organized a children's poetry club within the Japanese language division of the Sakai City Board of Education's elementary education research group. He recruited outside help, inviting the modernist poet Takenaka Iku, active from the prewar period, to hold a poetry study group with the children. Bessho had learned of the existence of *Giraffe (Kirin)*, Takenaka's poetry magazine for children, which had published its inaugural issue in February 1948. Bessho and three other teachers from nearby schools solicited children's works. By July of 1951, the four teachers had selected 25 poems and mimeographed 70 copies of *A Collection of Children's Poetry (Jidō shishū)*. They delivered copies to 28 elementary schools and the Board of Education in Sakai. One of the collected poems was "Outage" (*Teiden*). "Outage" was written by Tomo Fusako, a fifth-grade student at Minato Elementary School, one of the schools Bessho oversaw in his poetry group. The poem was short:

| | |
|---|---|
| Power outage tonight | *teiden no yoru* |
| And up there | *anna tokoro ni* |
| Holes in the tin roof | *totan no ana* |
| Like stars. | *hoshi no yō da* |

Bessho asked modernist poet Anzai Fuyue to give the magazine a more poetic title. Anzai christened the journal *Dove Flute* (*Hatobue*). Bessho published the second issue under this name in September. In the third issue, published in November 1951, Tomo Fusako's poem "Firewood Gathering" (*Shiba hiroi*) was featured in the second special selections section.

Bessho Yasoji published *A Collection of Minato Poetry* (*Minato shishū*) through the Japan Children's Poetry Study Group in June 1953, with Takenaka Iku providing a preface. Bessho placed Tomo Fusako's "Outage" at the top of the volume, and selected another eleven

poems for the collection: "Firewood Gathering", "The Bank" (*Ginkō*), "New Year's Day" (*Oshōgatsu*), "Mother" (*Haha*), "Swimming" (*Suiei*), "The Net" (*Ami*), "Glass" (*Garasu*), "I Always Think" (*Itsudemo omou*), "Electrician" (*Denkiya*), "Fisherman" (*Ryōshi*), and "Wish" (*Hoshii*).

Two of Tomo Fusako's poems—"The Bank" and "Fisherman"—exemplify the style and subjects that characterize her childhood works. Each would go on to be featured across a variety of poetry collections. "The Bank" reads:

| | |
|---|---|
| I went to the market with my mother and younger sister. | *Haha to imōto to san-nin de ichiba e itta* |
| We passed by the bank on the street corner. | *Ginkō no kado o tōtta* |
| "Mom, I've never | *"Okā-chan, koko e* |
| been in there", | *haittakoto nainā"* |
| my sister said. | *to, imōto ga itta* |
| I looked at my mother's face. | *Haha no kao o mitara* |
| She was silent. | *damatte ita* |
| I've never been inside, either. | *Watashi mo haitta kotoga nai* |

The poem lays bare the family's financial situation. Not only are they without a bank account, they have no connection to banking whatsoever. "Fisherman" illustrates the reasons for such a life and gives Tomo's vision for the future.

| | |
|---|---|
| Tacchan came by and is speaking with my father. | *Tacchan ga kite, chichi to hanashi o shiteiru* |
| "If these westerly winds keep up | *"Korekara nishikaze ga tsuzuitara* |
| my little boat is useless. | *chiisai fune yattara akan* |
| That's why I went to the factory". | *sonai omōte kojo e ittan'ya"* |
| My father is nodding and listening. | *Chichi wa "un un" to kiite iru* |
| There are fewer and fewer fishermen on Dejima | *Dejima wa ryōshi ga dandan hette iku* |
| and my mother and brother hate the work. | *haha mo ani mo ryōshi o iyagaru* |
| My father catches rides on other boats, | *Yoso no fune ni nosete moratteru chichi* |
| but when the west wind kicks up, or it rains, | *Nishikaze no tsuyoi hi ame no hi wa* |
| he can't go into the bay. | *Oki e ikarenai* |
| The river runs blue, | *Kawani nagarete iru aoi mizu* |
| purple, | *Murasaki no mizu* |
| red. | *akai mizu* |
| It's all water from the factory. | *Minna kōjō kara nagareru mizu da* |
| It's the water that kills the fish | *Sakana o korosu mizu da* |
| And not only in Sakai. | *Sakai dake dewa nai* |
| Osaka is filled with these factories. | *Osaka niwa takusan kōjō ga aru* |
| In three years | *Mō san-nen shitara* |
| I will go to the factory to work. | *watashi wa kōjō e itte hataraku* |
| Even on rainy days, | *Ameno hi mo* |
| even on windy days. I'll never take a day off. | *kazeno hi mo yasumazu* |
| I'll work hard. | *isshō kenmei hataraku noda* |

Tomo's poem opens by describing another local fisherman, nicknamed "Tacchan", who has decided to find work at a factory because he cannot make a living with his small boat. Tomo's own father is a hired fisherman without a boat of his own. Tomo calls out the cause of their impoverishment directly: the blue, purple, and red water that kills the fish. Pollution has made it impossible for small-scale fishermen to make a living in the coastal fishing industry. Tomo, however, plans to work hard at the factory to support her family after she graduates from junior high school in three years. Bessho quotes "Fisherman" in his afterword to the volume "With the Children" and explains it as follows:

> Fusako is a little housewife. She often goes absent from school to help with household chores. Yet she never loses her cheerfulness even as she struggles to babysit her younger brother and sister, cook for the family, and take care of her sick mother. She performs her chores willingly. Despite the terrible circumstances of her family's Dejima-based fishing business, her attitude towards life allows her to look forward to working outside of the home after graduating from junior high school. She has told me, "There are fewer and fewer fishermen on Dejima,

and my mother and brother don't want to fish for a living". A nearby factory is ejecting polluted water, leaving the Dejima fishing industry to harvest nothing more than small clams and shellfish. Because of its proximity to the big city, the people of Dejima are left to work as peddlers or laborers, rather than fishermen, or to earn a living through migrant labor. The industrial development in this area is clearly not compatible with the fishing industry. Fusako sees the writing on the wall and anticipates working in a factory. Fusako laments, "It's all that water from the factory. It's the water that kills the fish". Does her voice not carry the sorrowful cry of the people of Dejima? (pp. 122–23)

Tomo Fusako's "Firewood Gathering" and "Shelling Clams" (*Kai muki*) appeared as specially selected poems to open the January 1952 issue of Takenaka's poetry magazine *Giraffe* 5(1). Takenaka commented, "Both poems honestly express the hardships of life. Fusako's courage is precious. Things so often exaggerated are here written simply, stirring the reader's heart". Takenaka would eventually publish many of Fusako's poems in the pages of *Giraffe* from January to October 1952 and later contribute the essay "A Few Poems by Children" (Takenaka 1955). In this essay, Takenaka addresses Tomo Fusako's poems "I Always Think" and "Outage", writing, "This child is unstained by her poverty. These poems show both feet planted in a carefree life". He goes on:

> . . .I later had the opportunity to interview young Fusako. As expected, she was clever. Her responses to my questions were prompt and she never attempted to put on airs the way you find with children from an intellectual upbringing. (p. 234)

Takenaka's advocacy for Tomo's works continued. He wrote a short commentary on the poems "I Always Think", "New Year's Day", and "Outage" in *The Children are Watching: Life Captured in Poetry* (Takenaka 1959). From that small, mimeographed journal in Sakai, Tomo Fusako came to be highly regarded as an excellent poet, not only by her homeroom teacher Bessho and the other teachers editing *Dove Flute*, but also by professional poets such as Takenaka Iku.

Over the course of the decade, Tomo Fusako's reputation in the world of children's poetry grew. Her poems "Firewood Gathering" and "Shelling Clams" were included in *All-Japan Children's Poetry Collection* (Momota 1952). "Glass" was selected as one of the four special winners at the Children's Poetry Contest held on 22 November 1952, at the auditorium of Sakai Municipal High School. "Firewood Gathering" and "Teacher" (*Sensei*) were selected for *Japanese Children's Poetry of Daily Life* (Imamura 1953). *All-Japan Children's Poetry Anthology*, edited by Japan Children's Poetry Study Group, featured seven of Tomo's poems for their chapter of sixth-grade student poetry (Nihon Dōshi Kenkyūkai 1955). Educator and author Kumei Tsugane selected "The Bank" for the section "Poems on Daily Life" in his book *Poetry Classroom* (Kumei 1957). "Fisherman" was included in *New Class Structures: Six Years of Lectures on Classroom Management* (Katsuta 1957). Bessho was surely aware of this reach. The September 1957 volume of *Dove Flute* featured an article entitled "*Dove Flute*'s Poetry in Textbooks". This article announced that Tomo Fusako's "Outage" would be included in the fifth-grade Japanese language textbook edited by Yamamoto Yūzō, starting in 1958.

San'ichi Shobō, a publishing company specializing in poetry and other works produced by labor union literary circles at that time, became aware of Tomo Fusako and decided to publish a collection of her poems alongside other poetry and essays published in *Dove Flute*. Bessho acted as editor for two volumes published by San'ichi Shobō in 1958: *Red Clams: Tomo Fusako's Collected Poetry* (*Akagai: Tomo Fusako shishū*) and *Dove Flute: Poetry and Compositions by Elementary School Students* (*Hatobue: shogakusei no shi to tsuzurikata*). The former volume contains an autobiographical text by Tomo Fusako titled "My Early Years" (*Watashi no oitachi*) and a biography of the author. According to these original essays, Tomo Fusako graduated from elementary school in 1953, and from junior high school in 1956, and went on to work in a textile mill. She fell ill in 1954 and began working a temporary

position at a rubber factory. At the time of writing this essay, she had just left the company after three months due to layoffs and had joined another factory as a temporary worker.

The two volumes published by San'ichi Shobō furthered the reach of Tomo's works, and her poetry and essays continued to appear in edited volumes through the 1960s. Her sixth-grade composition "Father Goes to the Factory" (*Chichi ga kōjō e*), fifth-grade composition "Mother's Illness" (*Haha no byōki*), and fifth-grade poem "The Bank", were all reprinted from *Red Clams* in prolific author and educator Namekawa Michio's *An Anthology of Compositions from Boys and Girls* (Namekawa 1959). Japan Composition Asscociation included "The Bank" in their collection for the fifth grade (Nihon Sakubun no Kai 1962a), and "Fisherman" for the sixth grade (Nihon Sakubun no Kai 1962b). The January 1962 volume of the magazine *Fun Fifth-Graders* (*Tanoshii go-nensei*) reprinted "New Year" as if it were a poem written by a fifth-grade student at that time. Namekawa Michio's later edited volume also reprinted "Father Goes to the Factory", "Mother's Illness", and "The Bank" (Namekawa 1968). Amidst all of this publishing activity, the only confirmed writing by Tomo Fusako from this period was a poem titled "First Love" (*Hatsu koi*) in the February 1959 magazine *Life's Notebook* (*Jinsei techō*). Tomo submitted this poem as a seventeen-year-old factory worker. The poem expressed her feeling of lost love over a boy dressed in a school uniform and square hat, but the poem never received the same attention as her childhood works. The only "masterpieces" that remain are from her time as a poor schoolgirl penning beautiful examples of children's poetry.

### 3. From One Girl's Poem to Radio, Stage, and Television

The mainstream success of films such as 1937's *The Composition Class* and 1952's *School of Echoes* demonstrated to the Japanese film and broadcast industries that successful adaptations of literary works need not be restricted to famous authors; children's compositions were also viable source materials. Publishers also recognized this shift. The *Yomiuri Shimbun,* the national newspaper with the largest circulation in Japan, established a national composition contest for elementary and junior high school students and published collected editions of the best submissions.

On 9 January 1954, Asahi Broadcasting, an Osaka-based radio station, began broadcasting "My Free Poetry" (*Watashi no jiyūshi*), a program highlighting children's poetry. In December 1955, they added a composition segment to the program and changed the title to "Pencil Club" (*Enpitsu kurabu*), which went on to have a broadcast run of more than four years. In 1957, in the middle of the program's run, Asahi Broadcasting edited a volume of collected works selected from the total of more than 1500 pieces of composition and poetry submitted for broadcast. This volume, published by San'yōsha, shares a title with the original radio broadcast: *Pencil Club*.

NHK Television—Japan's national television station—broadcast "Composition Theater" (*Tsuzurikata gekijō*), a drama series based on compositions by elementary and junior high school students, from 1955 to 1958. Fuji Television also broadcast a drama series based on children's compositions and non-fiction bestsellers: "Fuji Home Theater" (*Fuji hōmu gekijō*) from 1960 to 1961. Tokyo Broadcasting System Television (also known as TBS) joined this boom, broadcasting a TV drama titled *Setsu-chan* on 22 January 1960. This drama was based on two compositions by Takaku Megumi, a third-grade student at Japan Women's University Elementary School: "Setsu-chan" and "Report Cards on the Heart" (*Shinzō no tsūshinbo*). The television production division of the famous film studio Nikkatsu also developed a serial drama entitled *Setsu-chan* for broadcast on Tokyo Channel 12 in 1964. Japan Education TV (now TV Asahi) broadcast *I Don't Want Mommy to Die* (*Kā-chan shigu no iyada*) on 4 December 1960. This program dramatized a best-selling essay by Hirabayashi Yoshitaka, an elementary school student. TBS aired a competing drama just weeks later on 23 December. Kansai Television broadcast *Kōjiro Box* (*Kōjiro bako*) on 31 December 1961, drawing from a prize-winning work in a composition contest. Mainichi Broadcasting System (MBS) Radio broadcast the radio drama *Ebara Radio Theater: A Bridge of Rainbows* (*Ebara rajio gekijō: niji no hashi*) on 22 February 1962.[2] This radio drama was based on the

book edited by Magawa Seita (Magawa 1961). NHK Television broadcast *TV Constitution* (*Terebi kenpō*) on 1 February 1962, based on a composition by Kayano Masaru, a sixth-grade student at Kanai Elementary School on Sado Island, Niigata Prefecture. Kayano had won both First Prize and the Education Minister's Prize in the 1960 Yomiuri Composition Contest. The script was written by Inoue Hisashi—who would go on to become a famous author in his own right—as a special feature to celebrate the channel's ninth anniversary. In 1963, NHK Television broadcast a series called "Children's Theater" (*Kodomo gekijō*), which also included original stories written by children. On 3 December 1965, Asahi Broadcasting aired the drama *The Love and Heart Series: The Tomorrow of These Children, Too* (*Ai to kokoro no shirīzu: Kono kora nimo asu ga*). This drama was based on the book *Composition Siblings* (*Tsuzurikata kyōdai*) by Nogami Tanji, Nogami Yoko, and Nogami Fusao, which had been adapted to Hisamatsu Seiji's *The Child Writers* in 1958.[3]

Several factors may have contributed to the popularity of these dramas based on children's poems and compositions. First, this period is roughly coeval with the postwar baby boomers entering elementary and junior high schools. As media consumers, the number of children growing into radio listeners, television viewers, and film audiences was substantial. Second, parents could be expected to bring their children to movie theaters that were showing films understood to be popular for (or inspired by) the baby boomer generation. Finally, the fact that these children, as the original authors, were paid little to no fees or royalties would have made for easier profits for the broadcasters and production companies.

It was within this media atmosphere that NHK Radio broadcast *Holes in the Tin Roof Like Stars* (*Totan no Ana wa Hoshi no yo*), a radio drama based on Tomo Fusako's "Outage" and other works, on 21 February 1958[4]. The scenario was written by Fujimoto Giichi, who was, at that time, still a student in Osaka Prefectural University's Faculty of Economics. Fujimoto was born in 1933 and entered the Faculty of Law at Ritsumeikan University in 1950, but withdrew because he was unable to afford tuition. He entered the Faculty of Education at Osaka Prefectural University, Naniwa in 1951 and transferred to the Faculty of Economics the following year. From 1955, Fujimoto repeatedly entered and won radio drama script contests. His fame as a scriptwriter grew at roughly the same time as Inoue Hisashi, who was, at that time, a student at Sophia University and also submitting scripts for competitions. In January 1958, Hashimoto Tadao, head of the production department of NHK's Osaka Broadcasting Station, showed Fujimoto a copy of Tomo Fusako's "Outage" in *Giraffe* and recommended that he use the poem to draft a radio drama script to air in their Friday drama slot[5]. Fujimoto and Hashimoto went to Tomo Fusako's house to conduct interviews, while Fujimoto later visited Bessho Yasoji's house and additional locations alone for twenty days in order to draft the script.

The radio drama opens with a monologue narration by Hatsuko, the drama's main character and stand-in for Fusako, reciting "Outage". Hatsuko explains Dejima's precarious situation. Chemicals are flowing out of a nearby factory, killing the plankton in the sea, and causing the fish to abandon the area. She describes her current work, removing the meat from red clams that have been trucked in from Kyushu, and she recites the poem "Shelling Clams". The voices of the other family members soon fade into the scene. News of a suicide in the neighborhood interrupts the family's discussion, though they later learn that the attempt failed. Hatsuko's mother is ill, and her condition worsens. Hatsuko recites a poem about mother's illness that is not in the original poetry collection. Hatsuko's family receives a laboratory report on the condition of the local water, confirming their initial fears: chemical pollutants have wiped out the plankton population. Hatsuko's father reads her poem "The Bank", describing the truth that she has never been inside a bank she walks by with her mother. Although Hatsuko's mother and father are embarrassed by this fact, her father admits that he too has never been inside a bank. Her father then sets out for the beach to give fishing one final try. He casts his net into the sea three times, but only hauls in two prawns.

In order to pay off his debt of JPY 30,000, Hatsuko's father decides to sell his boat and fishing equipment and take a job working as a guard for the local dyeing company—the same factory that discharges polluted wastewater into the ocean near Dejima. During yet another power outage that evening, Hatsuko and her younger brother look up at the holes in the tin roof before bed. It is here that she recites the titular poem "Outage". The next morning, her older brother, who was commuting to his factory job, suddenly returns home. He reports that a teacher from their school stopped him to announce that Hatsuko's poem "Outage" would appear in the school's textbook. The radio drama ends with Hatsuko, who has taken time off from attending school to shoulder more household labor (and clam shelling) from her ill mother, reciting the second half of the poem "Fisherman": "In another year / I will go to the factory to work", changing the original text's "three years" to just one.

Looking back on the drama in 1989, Fujimoto Giichi recalls how the script was ahead of its time: "To this day, I remain proud that *Holes in the Tin Roof Like Stars* is probably the first radio drama in Japan to address the subject of pollution ([Fujimoto 1989])". The late 1950s to the early 1960s witnessed the emergence of a social consciousness surrounding the so-called "four major pollution diseases" (*yondai kōgai byō*): itai-itai disease (a kind of cadmium poisoning in Toyama Prefecture) of 1955; Minamata disease in Kumamoto Prefecture in 1956; a second case of Minamata disease in Niigata Prefecture in 1965; and Yokkaichi asthma in Mie Prefecture, caused by sulfur dioxide. It was only with the enactment of the Environment Pollution Prevention Act in 1967 that "pollution" become a problem addressed by the government. Fujimoto's radio drama, which predates this law by about nine years, places a very early focus on the victims of pollution.

Prior to the original broadcast of the radio drama, the February 1958 issue of *Dove Flute* carried a feature article that declared to readers: "*Dove Flute* poetry to be broadcast as radio drama". The article requested, "Everyone, please tune in. Listen with your family, your fathers and your mothers". The following issue in March asks, "Did you listen to the broadcast, everyone? NHK is holding a contest for broadcast plays, so please fill out a postcard and vote for our play. If this drama wins first prize, we will award a prize to a selected voter". The next issue, in April, reported on the first page: "Thank you, everyone! *Holes in the Tin Roof Like Stars* has won first place!" This was accompanied by an announcement that the program would soon be rebroadcast. The rebroadcast actually occurred on 21 March 1958, which suggests that the April issue of *Dove Flute*, like most Japanese magazines, was already released by mid-March. Fujimoto graduated from Osaka Prefectural University in March of this same year. The radio drama's script was published in the April 1958 issue of the magazine *Broadcasting Culture* (*Hōsō bunka*). A note reads, "*Holes in the Tin Roof Like Stars* by Fujimoto Giichi has won the February edition of NHK Broadcasting's Friday Drama Competition. This award was decided by votes from listeners. Fujimoto's drama received 1730 votes out of 2038 total votes. The strength of *Dove Flute*'s organizational vote was likely a major factor in the drama's award, securing nearly 85 percent of the overall vote".

The May 1958 issue of *Dove Flute* carried another announcement that "*Dove Flute*'s poetry will be broadcast on TV".

> Have you all read Tomo Fusako's poetry collection *Red Clams*? Have you heard the radio drama it inspired, *Holes in the Tin Roof Like Stars*? *Red Clams* has also appeared on programs broadcast by ABC and Radio Tokyo. Now, OTV [Osaka Television] has decided to broadcast the poems on TV. They will appear for fifteen minutes on Sunday, 4 May, from 10:30 a.m. to 10:45 a.m. The program is called "Literature Here and There, Part Nine", produced by OTV Film. Please tune in!

Crosschecking against Osaka Television's program listings shows that "Literature Here and There" (*Bungaku no tokorodokoro*) aired every Sunday morning from 9 March to 8 June 1958, with reruns on Friday evenings from 25 July to 15 August and Sunday mornings from 24 August to November 30, though not in the order in which the programs original aired. The program featured works by famous authors set in the Kansai region. Examples include Yamasaki Toyoko's "The Shop Curtain" (*Noren*), Oda Sakunosuke's "Stories of

Osaka Life" (*Meoto zenzai*), and Hōjō Hideji's "The King" (*Ōshō*). This is the company that Tomo Fusako's "Red Clams" kept. The episode featuring Tomo's poetry was rebroadcast on Friday, 1 August at 6 p.m. and Sunday, 16 November at 11 a.m. (Kawasaki 2016).

On 14 September 1958, the theater group Ant Association (*Ari no kai*) staged *Holes in the Tin Roof Like Stars* (written and directed by Fujimoto Giichi) at the Ōtemae Theater (*Ōtemae kaikan*) in Osaka (Anonymous 1959). The Osaka Youth Affairs Council staged another production in 1959 for the theater company Kansai Geijutsuza. The company staged the play in their studio and also took the show to other stages across Osaka.

The play version of *Holes in the Tin Roof Like Stars* consists of three acts and four scenes. Act One opens with a scene set in Hatsuko's home. This tin-roofed fishing shack is packed with the family breaking open shells. A song is being hummed in the background, and a narrator sets the scene. Hatsuko is telling her brother a modified version of the Cinderella story, but her father slices his finger open with a knife while trying to rush everyone to finish their work. The family complains about their work shelling clams and Hatsuko's mother begins feeling ill and lies down. Hatsuko then begins to write a poem. A fellow Dejima fisherman appears and tries to kill himself with a razor, after which Hatsuko's mother, who has called for help, truly collapses.

Act Two begins the next day. Hatsuko's mother is sleeping; her younger brother brings a book of poems home from school that his teacher has entrusted to him; and her father reads "The Bank". The fishery union's laboratory report confirms the plankton die-off, just as in the radio drama. In the stage play, however, the father declares he will go out to cast his nets and stomps on Hatsuko's poetry book. Scene Two occurs in the pitch blackness of that evening, on account of the power outage. Hatsuko recites "Outage", and her father, who only caught two prawns, confesses that he will sell his boat and nets and get a job as a guard at the dyeing factory.

Act Three occurs the following morning. Hatsuko reads the second half of "Fisherman". The version here retains the same alteration from the radio drama: reducing the number of years until Hatsuko will go to work in the factory from three to one. Her older brother arrives with news that her poetry will appear in the school textbook. Hatsuko prepares to go to school for the first time in ages but changes her plans when a truckload of clams arrives for shelling. The curtain falls as Hatsuko gives her mother a break and begins working.

Writer and theater critic Shimizu Saburō criticized Kansai Geijutsuza's casting of the play (Shimizu 1960). But Hata Hisao, who played Hatsuko's father, was awarded the Osaka Nichinichi Shingeki Award for Best Performance by a Male Actor (Anonymous 1960a). Perhaps because of this award, *Holes in the Tin Roof Like Stars* was adapted into a television drama. This television adaptation was first broadcast by Fuji Television on 5 May 1960 at 3:30 p.m. and featured a cast drawing from Kansai Geijutsuza, including Hata Hisao, Shin'ya Eiko, and other members (Anonymous 1960b).

Due to its many appearances across a wide swath of mass media, *Holes in the Tin Roof Like Stars* has been a frequent source of quotations and plagiarism. In June 1958, novelist Mishima Shōroku published *Botchan of Asakusa* (Asakusa no Botchan). When Yuki, the heroine of the novel, looks out of her bedroom window at the night sky, she thinks, "'Holes in the Tin Roof/Like Stars' . . . Tonight reminds me of the poem I read one day in the newspaper by a poor girl". This is taken directly from "Outage". The haiku poet Shibata Hakuyōjo selected a poem by Ishidate Mitsuko of Wakayama as an "Honorable Mention" in the January 1961 issue of *The Schoolgirl's Friend* (*Jogakusei no tomo*). The poem includes the line, "The holes in my home's tin roof are like stars". Similarly, poet Akagi Kensuke selected a poem by Hosoya Kenji, a seventeen-year-old student from Gunma Prefecture, to appear in the December 1963 issue of *Life's Notebook* (*Jinsei techō*). The poem was titled "Tin Star" and began, "Holes in the tin roof, like stars". This is clearly based on either Tomo's original "Outage" or some version of *Holes in the Tin Roof Like Stars*. Akagi commented on the poem, saying, "It is a strange poem, almost like a nursery rhyme, but it is interesting

because it has a brightness that is undaunted by poverty". These editors do not appear to be familiar with the original poem.

The stage play *Holes in the Tin Roof Like Stars* was performed consistently and across the country during the 1960s. On 8–9 May 1962, a special performance was staged at the Kosei Nenkin Hall, directed by Mikage Shintarō (Anonymous 1962b). According to the "Theater News" column in the July 1962 issue of the theater magazine *Tragicomedy* (*Higeki kigeki*), this stage production was a "special performance for middle and high school students" and toured schools in Saitama, Kanagawa, and western Tokyo (Anonymous 1962a). The Sapporo Theater Company in Hokkaido toured with a production of the play in June 1963 under the name "Touring Theater". They visited Ashibetsu, Obihiro, and Kushiro over the course of about 20 days. These productions appear to have been well received (Hokkaidō Shimbun 1963). The company extended performances and had performed in Asahikawa, Fukagawa, Takikawa, Sunagawa, Iwamizawa, Sapporo, Muroran, Tomakomai, Noboribetsu, and Otaru by July of the following year (Hokkaidō Shimbun 1964).

These many media dramas and stage productions were all born from a short poem by an elementary school student. What began on the radio eventually drew wide attention, becoming a beloved production for radio, television, and theater over a comparatively long period. Certainly, one reason for this, in addition to the evocative power of the original poem, was Fujimoto Giichi's deft skill in adapting the poem—and its context—into an original script. I would, therefore, like to consider other works by Fujimoto to see how *Holes in the Tin Roof Like Stars* fits into his body of work.

## 4. Fujimoto Giichi and the Mass Media

After finding success in radio dramas and stage plays, Fujimoto Giichi tried his hand at writing for television in 1957. In 1958, director Kinugasa Teinosuke heard a radio drama performed in the Osaka dialect that Fujimoto had written for NHK. Kinugasa requested that Fujimoto rewrite one of Yasumi Toshio scripts that had originally been written in the standard Japanese dialect. Fujimoto's rewritten script, now in the Osaka dialect, was released as *Woman of Osaka (Osaka no onna)* by Daiei Tokyo in May 1958. Fujimoto was then asked to revise another Yasumi script into the Osaka dialect: director Kawashima Yūzō's adaptation of Yamasaki's "The Shop Curtain", mentioned above. This drama was released in June 1958 through Takarazuka Eiga/Tōhō. Fujimoto became an assistant scriptwriter at Takarazuka Eiga, eventually co-writing screenplays for Kimura Keigo's *Stray Cat (Nora neko)* in November 1958 and Yamazaki Yoshiteru's *The Man Who Came from the Sea (Umi kara kita otoko)* in March 1959. Fujimoto was then apprenticed to Kawashima, under whom he co-wrote the screenplay for *Room for Rent (Kashima ari)* in June 1959, an adaptation of an original story by Ibuse Masuji. Fujimoto left Takarakuza Eiga in 1964 but continued to work on many scripts. It was his television appearances, beginning in 1965, that transformed his fate.

In 1965, he was asked to appear as a guest on a new Yomiuri Television program, "11PM". Instead, he wound up as the host. He kept the role for the next 25 years, eventually stopping in 1990 (Kiyokawa 2013). Although the program consistently achieved high ratings, it was criticized in the mass media as "vulgar"—the controversy stemmed from the regular inclusion of nudity and other crude segments. The requests for Fujimoto to write scripts for NHK and commercial broadcasters dropped sharply (Fujimoto 2011). Fujimoto turned to novel writing in 1968. He was nominated for the Naoki Prize twice in 1969 and once in 1971, though he did not win. He finally won the Naoki Prize in 1974 for his novel *The Demon's Poem (Oni no uta)*, based on rakugo storyteller Katsura Beikyo II. In his commentary to this prize-winning text, literary critic Komatsu Shinroku speculated that the reason for Fujimoto's delay in winning the Naoki Prize may have been "a combination of bad luck for being the infamous host of '11PM'" (Komatsu 1976).

Looking at Fujimoto's career trajectory, it appears that being active in the mass media and achieving fame as a novelist were two contradictory aspects of his art. In the film adaptation genres that have long enjoyed a degree of popularity, i.e., the "literary film"

and "literary drama" genres, any number of highly respected authors gave permission for their original literary works to be adapted to film. This created a division of labor wherein the people involved with mass media worked on film and drama adaptations. Of course, there are counterexamples, such as Tanizaki Junichirō writing the script for Thomas Kurihara's 1920 "The Amateur Club" (*Amachua kurabu*) or Kawabata Yasunari writing the script for Kinugasa Teinosuke's 1926 "A Page of Madness" (*Kurutta ichi pēji*). These examples, however, come from when film was still a young medium. Abe Kōbō's deep involvement in the films of Teshigahara Hiroshi or the television dramas of Wada Ben remain rare examples. There are few examples of postwar writers who gained fame through films or dramas. Conversely, writers such as Morimura Seiichi, Tsutsui Yasutaka, and Akagawa Jirō, who became bestsellers through Kadokawa's "media mix" (as discussed by Zahlten), were originally active in entertainment works and tended to be dismissed by the literary world. Fujimoto Giichi was no exception to this trend.

Tomo Fusako, who became famous as a child poet and writer thanks in part to Fujimoto's *Holes in the Tin Roof Like Stars*, did, in fact, go on to factory work after graduating from junior high school. She worked at a woolen textile factory in Izumi-ōtsu, Osaka, but soon quit due to poor health. The job was far from her home and regularly required two hours of unpaid overtime in addition to her standard eight-hour workday. She became a temporary worker for Akiyama Rubber, but was laid off. She worked as a temporary worker for Shimano Industries, world famous for Shimano bicycle parts, but left because of seborrheic dermatitis. She took up dressmaking and typing while collecting unemployment benefits and became a telephone operator. She eventually married at age 21 and raised two sons.

It is, in fact, thanks to mass media that we know about her post-childhood life and career. Fujimoto wrote an article in the 11 September 2000 evening edition of the *Asahi Shimbun* titled, "My Professional Origins Trace to a Girl's Poem". Fujimoto quotes Tomo Fusako's "Outage" and "The Bank" from memory—and somewhat incorrectly. He describes his experience of first reading the poems as "an electric current running through my whole body" and how he adapted the poetry to the radio drama format. A response letter was published in the 18 September evening edition of the *Asahi Shimbun*. The author was a reader and a "Fujimoto fan", and submitted the following information: "That poem was written by Tomo Fusako, a fifth-grade elementary school student at the time. It first appeared in the inaugural issue of *Dove Flute*". Additional information appeared in the 2 October evening edition of *Asahi Shimbun*. This time, the principal of Sakai City's Ebaraji Elementary School—and head of the *Dove Flute* Association—Nakamura Kōji wrote in regarding the editing process underway for producing a 50th anniversary collection of *Dove Flute*. This letter included supplementary materials from *Dove Flute*'s editor-in-chief, Minami Hideki. An *Asahi Shimbun* reporter, Kawai Mamie, interviewed Tomo Fusako, Fujimoto Giichi, and others, and wrote an article in the 23 October evening edition titled, "The Poetry Girl, Now Living Haiku Days: Fujimoto Giichi's 'Origins'". This article featured Honjō Fusako, née Tomo, now 59 years old. "You remembered me? I'm so glad", she said, and explained her life after being a child poet.

> Later, I worked as a factory worker and telephone operator to support my family. I married at twenty-one and raised two sons. I never told my family that I had written poetry, nor that my poems had been adapted into a radio drama by Mr. Fujimoto. My poetry notebook and newspaper clippings are all gone now. I haven't written poetry since then.

She did, however, began writing haiku at the age of forty. "Poetry and haiku are the same", she said. "They allow me to forget the unpleasantries of daily life and concentrate on other things. Yes, I will continue to write haiku until I die". Fujimoto heard this and commented on Tomo's recent activities, "I see that she is writing again, now in the form of haiku. That's wonderful. Fusako's poems were originally like haiku. The rhythm and the ideas must be the same". Tomo Fusako continued to submit poetry to the Nara-Prefecture-based Shichiyō Haiku Association (*Shichiyō haiku kai*) and their magazine, *Shichiyō*. One

of her poems was republished in the October 1999 volume of Kadokawa Shoten's major poetry magazine, *Haiku*. The poem read:

| | |
|---|---|
| I gave the seed pack | *Tanebukuro* |
| a little shake and cradled | *futte midorigo* |
| my precious baby | *ayashikeri* |

This haiku detailed her day-to-day life with her grandchild. The final poems Tomo published in *Shichiyō*—appearing in the June 2008 volume—carried on this theme:

| | |
|---|---|
| Whispering voices | *Tanebukuro* |
| from tiny flowers when I | *fureba kobana no* |
| shake up the seed pack | *sasayakeri* |

These lines were published some fifty-seven years after "Outage". Although alive and well, Tomo Fusako no longer writes poetry.

We might place Tomo/Honjō Fusako in the historical context of other writers whose works had been adapted when they were still quite young. Toyoda Masako did not receive any royalty income after her writing was published in the 1937 bestseller *The Composition Class*. She trusted her teacher Ōki Ken'ichirō, however, and was eventually adopted by him. After Oki's death and a breakdown in her relationship with his widow, Toyoda revealed the details of her experience in *The Seedling* (Toyoda 1959). Adaptations based on children's poems and compositions existed as a way to exploit children who did not know their own rights. Certainly, Tomo Fusako/Honjo Fusako also belongs to this lineage. Her subsequent life, however, one where she was able to live happily with her family and write haiku, comes as a relief. In some sense, it brings closure to Fujimoto—and to us, the readers.

## 5. Conclusions

We might consider Tomo Fusako's relationship with media (and the media industry) as vacillating between two models. On the one hand, we might imagine her position as part of the "colonial fantasy" that Jacqueline Rose identified in literary works for and about children, where there is "a belief in childhood as something which is able to by-pass the imperfections of the civilized world" (Rose 1984). In a material sense, also, adults active in the *industry* of media production invaded and controlled Tomo's literary voice in such a way as to extract value from her original discourse. Film companies and broadcasters established a system wherein they could produce "original" works based on children's compositions (from Toyoda to Tomo) without ever providing proper compensation. On the other hand, as Clémentine Beauvais has responded to Rose, we can also see Tomo possessing a certain affective power over the adults creating and consuming this very same media by virtue of her voice as a child (Beauvais 2019). Here, we might point to Fujimoto's reciprocal creative relationship with Tomo, in particular. Beyond Tomo's original readership, Fujimoto effectively adapted Tomo's childlike diction and style to create radio dramas and plays. Through this generative process, Tomo exerted considerable influence over Fujimoto's creative output.

Despite being a widely known writer, Fujimoto Giichi's literature has not been the object of academic research. In a literary world centered on so-called "pure literature", Fujimoto's activities in the mass media relegated him to an outsider position, a writer whose works were not worthy of consideration. His novels and scripts, which do not fit neatly into any familiar category, such as pure literature, minority literature, nor even popular literature, fall outside the purview of traditional literary studies. By contrast, films and dramas based on children's writing have only rarely been addressed in previous studies. Aside from a few exceptions, the research tends to focus on smash hits or best-sellers, such as *The Composition Class*. As I have argued above, however, these other works deserve serious consideration—both in their own right as literary texts and in the space they occupy in the media industry of modern Japan.

**Funding:** This research was funded by the Japan Society for the Promotion of Science, Grant-in-Aid for Scientific Research (C), grant number 20K00302.

**Institutional Review Board Statement:** Not applicable.

**Informed Consent Statement:** Not applicable.

**Data Availability Statement:** *Dove Flute (Hatobue)* and related publications are available at the Sakai Municipal Library. Other journals and publications are available at the National Diet Library.

**Acknowledgments:** I would like to thank Eric Siercks for comments on a draft of this article.

**Conflicts of Interest:** The author declares no conflict of interest.

## Notes

1   I have already given an oral presentation as "Authoritative Gentleness around Colonial Children's Compositions: On Ch'oi Ingyu's Tuition (1940)," Cultures of Crossing: Transpacific and Inter-Asian Diaspora Symposium, University of Utah (& online), 4 December 2021, and have plans to make it into a paper.

2   A recording is available via The Broadcast Library: https://www.bpcj.or.jp/ (accessed on 17 October 2023).

3   The above examples are based on data from the TV Drama Database: http://www.tvdrama-db.com/ (accessed on 17 October 2023) and the Broadcast Library.

4   Two versions of scenario are available via Fujimoto Giichi Archive: http://fujimotogiichi.nkac.or.jp/ (accessed on 17 October 2023). A recording of the 1967 reproduction is available via Niconico: https://www.nicovideo.jp/watch/sm19301210 (accessed on 17 October 2023).

5   See (Fujimoto 1960, 1976). This seems to indicate that he first came into contact with the poem and then went to the production manager to ask about producing a radio dramatization, but his recollection at a point two years later seems more reliable.

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
