# Peer review of "Intersections of Children’s Poetry, Popular Literature, and Mass Media: Fujimoto Giichi’s Adaptation of Holes in the Tin Roof like Stars from Tomo Fusako’s Poem to Radio Drama"

_humanities, doi:10.3390/h12060128_

Round 1

Reviewer 1 Report

Comments and Suggestions for Authors

This was an absorbing, informative and at times even a moving article. It has value as a detailed factual account of the dissemination of a small set of texts across a number of different media and audiences in the landscape of 1950s and ‘60s Japan.

There are however various ways in which this piece could have been much stronger.

While the story is of intrinsic interest, it invites and could usefully have received some comparison with the situation in more recent decades. It would have been useful to have a firmer sense of whether, or how far, this was a typical trajectory.

More importantly, the article lacks a consistent focus, veering between Tomo and Fujimoto’s careers, but not really making a strong comparison between them or drawing any conclusion, even though the ingredients all seem to be available. This tale of two writers thwarted or at least held back might, for example, have been framed in terms of a comparison of the compound structural disadvantages suffered by Tomo as a) a child, b) poor and c) female, and those of Fujimoto, who was none of these things but was seen as in some sense ‘sullying’ his literary status by his work in less respectable media and genres – and the extent to which either’s situation was recoverable. (I find it strange but telling that only Fujimoto’s name is mentioned in the article title, although it is at least as much about Tomo.)

The related issues of exploitation (did Tomo, who needed it most, benefit materially at all from the multiple reworkings and quotations of her work and story?) and of the general cultural positioning of the ‘child author’, were mentioned, but in both cases rather glancingly; they would benefit from a more direct focus. For some useful background to this aspect of the reception of child poets, albeit in a Western context, I highly recommend Clémentine Beauvais’s ‘Is There a Text in This Child? Childness and the Child-Authored Text’, Children’s Literature in Education 50, 60–75 (2019).

Finally, I would have liked to have seen some quotation and analysis of some of the other poems frequently mentioned, rather than ‘Outage’ alone. So much of this article’s point depends on the reception of Tomo’s writing that this omission seems odd.

Comments on the Quality of English Language

The English is generally good. However, there were quite a few minor errors, such as omitted macrons or confused name orders (e.g. ‘Yuzo Yamamoto’ should be ‘Yamamoto Yūzō’, according to the conventions adopted elsewhere in the article). These should all be checked carefully.

Author Response

Thank you for your careful peer review and helpful comments.
I do not have an answer myself as I am still researching whether there are any recent examples. I believe that this example, while not typical, may be unique and worthy of consideration.
You are absolutely right in pointing out that Tomo Fusako should be mentioned and considered, so I have revised the title to include her name as well.
Thank you also for the instruction on Bauvais' paper. It was a very useful suggestion and deepened the discussion.
I also found and added other poems and haiku by Tomo Fusako, but it took me a long time to research the haiku in particular. Thank you for your patience and I hope it has turned into a good paper.

Reviewer 2 Report

Comments and Suggestions for Authors

I was fascinated by this article and thoroughly enjoyed reading it. It is organized well and I particularly appreciated the historical set up. I would have appreciated a bit more cultural information about that era though, as it seems important to explain--"why" was there such interest at that time in children's poetry? or programing about writing?

If I have one criticism it is this: The author should provide either some texts of the poems mentioned, or set up a website where they can be read, or give a short appendix to explain where and how to obtain these works by children. As I was reading the article I am googling the titles and author names and could not find anything. To me, part of this author's purpose is to promote the study of these fascinating works by children---yet where can I get them? How can I read them to further their study? 

The English in this essay is quite good. I did find 3 instances of a word missing that should be added:

Line 62, add "he" : he was a university student and "he" went on to....

Line 202: What is the Yomiuri Shimbun? Please explain what this is?

Line 506: needs "with"---"with" his widow

Author Response

Thank you for your careful peer review and helpful comments.
The question of why so much attention was paid to children's poetry is still under research and I do not yet have a concrete answer; I have a hypothesis that it may have something to do with the popularity of children's magazines from around the 1920s and the movement for writing in education, but I am not yet ready to write clearly. However, I have been able to reinforce my conclusions with suggestions from other peer review comments and by referring to Rose's and Beauvais's arguments.
After a long period of research after receiving the peer review comments, I was able to locate Tomo Fusako's current address and contact her by phone. However, I could not get permission from her to publish her poems on the Internet, as she is getting old. This time, I have included a few poems and haiku quotations to give readers an idea of what her poems are about. I hope you will understand.